# Application of the Decision-Making Trial and Evaluation Laboratory Method to Assess Factors Influencing the Development of Cycling Infrastructure in Cities

**Anna Trembecka [1], Grzegorz Ginda [2],* and Anita Kwartnik-Pruc [1]**

1    Faculty of Geo-Data Science, Geodesy, and Environmental Engineering, AGH University of Krakow, 30-059 Krakow, Poland; anna.trembecka@agh.edu.pl (A.T.); anita.kwartnik@agh.edu.pl (A.K.-P.)
2    Faculty of Management, AGH University of Krakow, 30-059 Krakow, Poland
\*    Correspondence: gginda@agh.edu.pl

**Abstract:** Sustainable development and environmental considerations have resulted in many cities around the world recognising the importance of non-motorised modes of transport. Problems related to the proper development and maintenance of cycling infrastructure have already been the subject of various studies. However, they have mainly dealt with the identification of factors influencing the development of cycle paths and the optimisation of the design of safe and comfortable cycle routes. The influence of individual factors on each other and on the development of cycling infrastructure has not been studied. The research aims of this article are to identify which factors influence the development of bicycle infrastructure, their role and interdependence, and their prioritisation. It also looks at whether there are differences between the opinions of bicycle users and experts professionally involved in the development of bicycle paths in assessing the importance of the factors indicated. As a result of the study, eight factors influencing the development of bicycle infrastructure were identified. Based on the opinions of cyclists and experts, the nature of each factor was analysed. Taking into account the complex relationships between the factors, the key factors contributing to the development of bicycle infrastructure were shown: (1) the planning of bicycle paths, taking into account the separation of individual paths and their continuity, consistency, and length; (2) legal regulations promoting cycling in terms of transportation policy; (3) the elimination of obstacles; and (4) the design of bicycle paths, taking into account the safety, space management, terrain, and attractiveness of the surroundings. The results for both groups of respondents were compared. They indicate that both groups of respondents reported the same factors as the most important, with the only differences being in the order of the importance of the factors. The academic value of this work lies in showing the usability of the underrated original version of DEMATEL methodology in the considered area for key factors. The practical significance of this paper is the provision of a rather simple, yet reliable, tool for addressing the complexity of interrelated issues that make the development of urban infrastructure a cumbersome task.

**Keywords:** bicycle paths; bicycle infrastructure; DEMATEL





## 1. Introduction

Sustainability and environmental concerns have led many cities around the world to recognise the importance of non-motorised means of transport to improve quality of life. Significant challenges are the proper maintenance and development of bicycle infrastructure for the safety and comfort of cyclists. As an important non-motorised means of transport, the bicycle is considered an efficient alternative to motorised vehicles. It contributes to reducing emissions and road congestion. Creating safe and comfortable bicycle routes has become one of the main development tasks of modern cities. More and more cities can boast of having a good cycling infrastructure and a commitment to promoting an ecological and sustainable means of transport, which is the bicycle.

The solutions implemented in urban transport systems in Western Europe in the field of bicycle policy are an illustration of the pro-ecological trends that are becoming more common, not only in transport but also in other areas of the economy. A major problem in cities in Poland is the lack of coherence of the bicycle network. This is caused by the lack of a comprehensive approach and a broad view of the issue of sustainable mobility, the essence of which is the creation of attractive alternatives to excessive car use [1]. Therefore, the development of bicycle paths and public transport should be combined with restrictions on cars and the construction of underground and multi-storey parking lots on the outskirts of city centres or at public transport hubs (park + ride). Bicycle paths are often built "on the occasion" of street renovations, which results in only sections of them being built rather than a connected network. Their maintenance is also a problem because they are not prioritised for cleaning or the removal of vegetation or snow. There are almost no solutions giving priority or an easier passage to cyclists at intersections that significantly facilitate cycling, e.g., as in Copenhagen. According to the author of [2], the most important barriers hindering the implementation of the bicycle policy in Poland include the following:

- The integration of bicycles with the existing transport system, most often based on buses and trams, as well as developing a solution enabling efficient and convenient changing from one means of transport to another.
- Topography and other natural obstacles that prevent or significantly hamper the construction of bicycle infrastructure.
- Society's attitude towards cycling; often, convincing people to change their means of transport requires a long-term promotional programme.

Despite the difficulties, the actions of the government and local authorities aimed at reducing the level of pollution in the city and increasing the share of bicycle traffic are worth emphasising. These actions are consistent with the provisions contained in the "Transport development strategy until 2020 (with a perspective until 2030)" developed by Poland's Ministry of Transport, Construction and Maritime Economy, which assumes the promotion of pedestrian and bicycle transport [1]. The cycling infrastructure in Poland is not at a sufficient level compared to that of the European leaders, e.g., the Dutch or the Danes [3], which is why we took up this topic in our research. According to data from the Central Statistical Office, in 2020, the total length of the bicycle paths in Poland managed by local government units was 17.3 thousand km. Compared to 2019, there were 1.7 thousand more km of bicycle paths (i.e., 11%) [3]. An increase in the density of the bicycle infrastructure network (the ratio of the length of the bicycle infrastructure to the city area) was also noted in the surveyed cities. According to the report [3], Polish cities develop various types of planning and strategic documents regarding bicycle infrastructure, but they lack indicators that would enable the monitoring of the assumed goals.

The survey was carried out in Krakow, the second largest city in Poland. In Krakow's development strategy, "I want to live here. Kraków 2030" [4], developed by local authorities, one of the strategic goals is "Kraków-resident-friendly, effective and ecological transport system". This goal includes the development of bicycle infrastructure, along with bicycle parking lots. The implementation of these goals is specified in the programme for the construction of bicycle paths [5]. In 2019, a study of the basic bicycle routes of the City of Krakow was developed [6]. As part of the document, an inventory of existing bicycle paths was carried out, including the verification of the implementation of the assumptions presented in the "Investment Programme—Study of Basic Cycle Routes" adopted by Resolution No. CIX/1493/10 of the Krakow City Council of 22 September 2010. The study [6] also identified problem areas on existing routes. The document presents a network of bicycle paths divided into main routes, connecting routes, and recreational routes, and it also includes an indicative estimate of the construction costs of the bicycle paths proposed in the document. For the purposes of the document, investment priorities were set, and planned sections of bicycle paths were assigned to them. Despite the actions taken by the authorities, the bicycle paths in Krakow still do not constitute a coherent network. Many sections are missing (Figure 1a), and others require renovation or redesign

(Figure 1b). Other problems are pedestrian and bicycle paths (Figure 1c), which do not provide sufficient safety for pedestrians or cyclists, and bicycle lanes separated on one-way roads (Figure 1d), which are narrow and dangerous.

(**a**) no continuation of the bicycle path

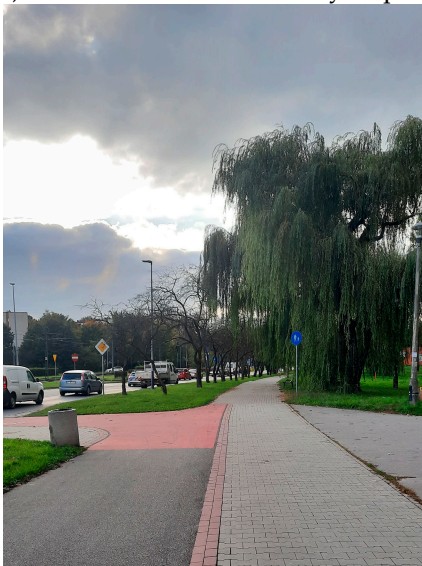

(**b**) bicycle path requiring renovation

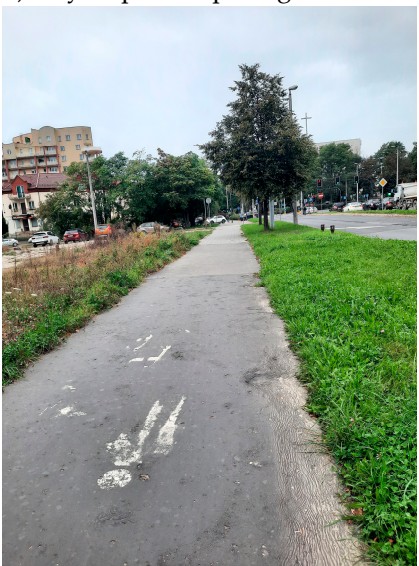

(**c**) combining the bicycle and pedestrian paths into one

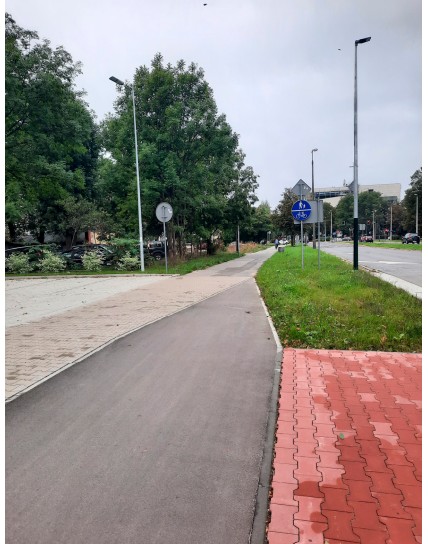

(**d**) bicycle lane separated on one-way road

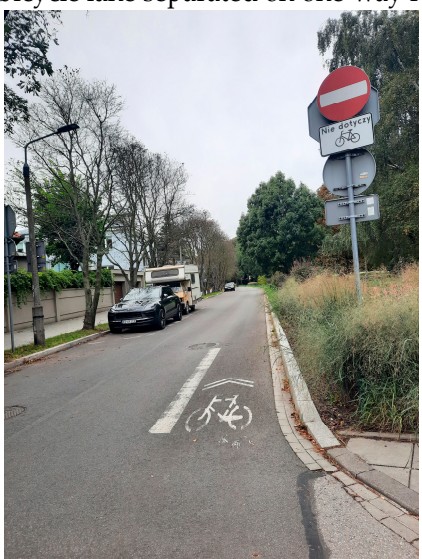

**Figure 1.** Illustration of problems that occur on bicycle paths in Krakow.

The development of bicycle paths is very important from the point of view of sustainable urban development. The research objective of this paper is to answer the following questions:

- Which factors influence the development of bicycle infrastructure?
- What is the role and interdependence of these factors in the analysed process?
- Which factors are prioritised in the development of bicycle infrastructure?
- Are there differences between the opinions of bicycle users and the experts professionally involved in the development of bicycle paths in assessing the importance of the factors indicated?

The Decision-Making Trial and Evaluation Laboratory (DEMATEL) method was chosen to achieve the research objective because of its maturity, flexibility, and extensively verified quality. This method allows for the efficient identification of all cause–effect relationships and key factors, even in very complex systems.

The DEMATEL method is based on expert knowledge, which is why the authors prepared a survey and asked 18 users of bicycle paths and 5 experts—representatives of public administrations professionally involved in planning, building, and promoting bicycle paths—to assess the impact of individual factors on each other on a five-point scale. Based on the survey results, the nature of the factors influencing the development of bicycle paths was analysed. The results from the surveys of the bicycle path users and experts professionally involved in the development of bicycle paths were considered separately. By using the DEMATEL method, which takes into account complex connections between factors, the key factors in the development of bicycle infrastructure were identified. The roles of other factors and the relationships between them were also determined. Differences in the results for both groups of respondents were also identified.

The remaining part of this paper is structured as follows: The second section is devoted to a review of the literature that covers bicycle paths and the application of the DEMATEL method. The third section addresses the methodology for the identification of key factors, discusses the input results, and details the DEMATEL technique procedure. The fourth section deals with the presentation of the obtained DEMATEL analysis results. The outcomes of the analysis are discussed in the fifth section. The last section is devoted to the final conclusions and plans for future research.

The results of the research can be put into practice in various countries for improving conditions that support cycling. They can also be useful for directing transport development strategies in terms of bicycle transportation in different regions and countries. The analyses carried out may inspire further scientific research on, among other things, the conditions affecting the design of bicycle routes in different countries.

## 2. Literature Review

The literature review is divided into three subsections. The first subsection provides a broad overview of the literature thematically related to the development of bicycle infrastructure. The second subsection regards the use of the DEMATEL method in the transport and infrastructure sectors. The last one covers the gap analysis and research highlights.

### 2.1. Issues Related to the Development of Bicycle Paths

One of the priorities of sustainable transport is the development of a multimodal transport system in which the planning of bicycle infrastructure plays an important role. The studies presented in [7] indicated that travel mode choice is affected by travel-related attitudes. A positive stance towards a certain travel mode increases the probability that people will choose that mode for a particular trip. The results indicate that about half of the respondents chose a non-preferred travel mode and that dissonant travellers could mainly be found among public transport users, whilst the least could be found among cyclists. This is partly due to the low positive stance among public transport users towards their mode of transport and the highly positive stance among cyclists towards their mode of transport. Transport service provision in many urban areas is dominated by car users, resulting in several traffic externality issues (e.g., noise, pollution, and accidents). The study in [8] investigated the perception and satisfaction of private vehicle (PV) users, including micro-mobility users, during their commute by car in a Central/Eastern European country context. The key findings indicate that PV users can be attracted to using sustainable transport by designing the travel service quality to provide the level of service desired by customers. The study in [9] investigated how socio-demographic variables affect the ratings of the inconvenience of daily commuting. The results indicate that people who travel actively (walking/biking) tend to be less dissatisfied with their commute, followed by those who travel in a personal vehicle and transit users. A number of attitudinal responses

are shown to impact the desire to travel more or less, including variables that relate to the social environment, the availability of local activities, the quality of facilities, the productive use of the commute, and the intrinsic value found in commute travel. Cycling is a very desirable form of transport in highly urbanised areas. It does not cause traffic jams, does not cause noise, and does not produce air pollution. In fact, cycling reduces air pollution because, to put it simply, each active cyclist is potentially one less car on the road [10]. It is a convenient, fast, and cheap way of getting around, which may ultimately contribute to changing a citizen's lifestyle to a more active one.

Apart from the basic function of moving people from one place to another, bicycle transport has many additional functions [11,12], including tourism, recreation, sports, and health. Due to the location of bicycle paths in cities, often in the vicinity of roads used by cars, the problems of polluted air and noise have also been the subject of research [13–18].

The main objective of one study [14] was to develop a model for planning and generating urban routes for active modes that have a less negative influence on active users regarding their exposure to air and noise pollution. In this research, the model used to assign the best route was presented to cyclists from an environmental point of view by defining the least polluted, least noisy, and most health-friendly path. The influence of noise, vibration, the presence of cycle paths, and the period of the day on the stress experienced by cyclists was analysed in [18]. In [19], problems limiting cycling due to cultural values and customs were identified. Negative aspects of individual behaviour can only be eliminated through cultural change, e.g., organising information campaigns to create social awareness in the field of sustainable development. Cycling infrastructure requires continuous expenditure, both for its promotion and modernisation, and the construction of new facilities to meet the various needs of users.

The major cities in Europe with the highest share of bicycles in road traffic are Amsterdam and Copenhagen. In the largest city in the Netherlands, 40% of all urban trips are made by bicycle, and, in the capital of Denmark, this is over 30% [20].

Appropriate infrastructure, transport policy, and public awareness are required to prioritise and encourage the use of bicycles over other motorised modes [21]. According to the author of [21], there are five indicators determining a city's readiness for cycling. These indicators are the city's population, the length of the road network, the city's form, the city's area, and the modal division of motorised transport. All of these metrics are positively correlated with cycling, except for city form, which is negatively correlated because linear city forms result in longer journeys that are more difficult to cycle. Research conducted in India [22] identified and assessed important bicycle-friendly conditions. Similar issues are presented in [23], which analysed the roles of bicycle parking, bicycle showers, free parking spaces, and transport benefits as determinants of cycling to work. Research in Maryland, United States, examined the associations of land use, built environment, demographics, socioeconomics, and traffic conditions with cycling [24].

Designing safe infrastructure for all categories of travellers, including cyclists, is becoming a basic requirement in the light of sustainable urban development [25–27]. Planning and policy efforts at all levels of transportation planning aim to increase bicycle transportation. In many cases, initiatives are motivated by the desire to reduce car use and the accompanying environmental consequences (e.g., pollution and the consumption of natural resources). Alternatively, they are motivated by public health concerns [28]. The literature presents various solutions supporting the design of safe bicycle infrastructure. A useful tool is the context-sensitive design approach. In this way, it is possible to examine a project or an existing road, reporting its failure potential and safety status and detecting its shortcomings, taking into account the communes and areas through which it runs [25]. Another method to support the planning of bicycle facilities proposed in [29] is the use of a multi-criteria analysis (MCE) to integrate criteria based on supply and demand in the planning of bicycle infrastructure. In the study, the analysis was performed at two geographic levels, namely, the network level and the neighbourhood level, and an exploratory spatial data analysis (ESDA) method based on a geographic information system (GIS) was used to examine the

spatial patterns of bicycle facilities at the neighbourhood level. The results suggest that combining GIS and MCE analyses may serve as a better alternative for planning optimal bicycle facilities. Another paper [30] investigated which parameters have the highest influence on cyclists' route choice behaviour and how they contribute. The model used for creating the bicycle route choice programme is based on the network model of Norrköping (Sweden). For comparing the results of the cost function and the shortest route (between an origin and destination), the model has a shortest path finding algorithm between different origin and destination pairs. In [31], an optimisation framework was proposed for urban bicycle network design. The model takes into account the interests of the users (who travel along the shortest paths) and the planners (the available budget). Creating a network of bicycle paths for underdeveloped areas is problematic due to the fact that the degree of their use is unknown. The authors of [32] proposed a method for determining a theoretical network of routes connecting objects with the highest concentration with the shortest routes using the urban road network. The method allows for the preliminary planning of bicycle infrastructure for areas with low levels of development. Proposals for the planning of bicycle paths based on actual bicycle trajectories and stated-route-choice studies are presented in [26,33,34]. The author of [35] analysed the diversity of commuting to work by bicycle in large American cities, with particular emphasis on assessing the impact of bicycle paths and lanes on the increase in bicycle traffic in the United States.

The construction of bicycle paths has become a key task for governments promoting healthy lifestyles [36]. Well-planned cycle paths can reduce traffic congestion and reduce safety risks for both cyclists and motor vehicle drivers. A significant subject of research has been the interaction of cyclists with other road users in the context of various infrastructure solutions, as well as in relation to potential conflict situations [36–40] or those resulting in accidents [41,42]. The best way to reduce or eliminate conflicts is to designate speed limit zones and increase the number of bicycle lanes [43–45] and separate bicycle infrastructure [46]. The relationship between safety and the condition of bicycle infrastructure was also examined. The technical condition of a bicycle path and the width of bicycle paths influence the behaviour of their users and their tendency to take greater risks [47–49]. Cyclists are more careful and reduce their speed on paths shared with pedestrians. On paths designated only for bicycles, they move at higher speeds while maintaining the same level of safety [50]. Conflicts between cyclists and other road users may therefore result from the organisation of infrastructure and occur depending on the width of the road, the designated direction of traffic [51], and the space shared with pedestrians [52].

One of the new, innovative elements of bicycle infrastructure is the bicycle rental system, which is being introduced in an increasing number of countries. Bicycle sharing, due to its zero emissions and shared nature, is perceived as one of the sustainable means of urban transport [53,54]. Such a system requires self-service and renting and operates in different parts of urban space, often with the help of special docking-equipped stations with a customer service terminal [55,56]. The intended use of shared bikes is usually aimed at offering bikes for rent for short periods and distances [57,58]. The economic benefits provided by the bike-sharing system are primarily related to shortening travel time, especially in large agglomerations [59]. The launch of the innovative Nextbike bicycle rental system in Warsaw resulted in an increase in interest in bicycle rental [60]. As a result, several other Polish cities (e.g., Poznań) launched competitive bicycle rentals. Some rental companies, thanks to cooperation with other urban means of transport (e.g., buses, trams, and light rail), significantly improve transport efficiency [61,62], mainly in the central parts of cities [63]. Equally important is bicycle parking. A nationwide programme to modernise regular and secure bicycle parking at Dutch railway stations has led to increased user satisfaction and an increase in the number of bicycles parked at stations [64].

We also cannot forget about countries where the use of bicycles as a means of transport is still very low and road infrastructure for cyclists is practically non-existent [65]. Here, the research focuses on the authorities' actions in implementing cycling policy at individual national, regional, and local levels, depending on the level of the problems that need to be solved.

### 2.2. The Use of the DEMATEL Method in Transport Infrastructure Sector Research

DEMATEL (Decision-Making Trial and Evaluation Laboratory) is an established, well-developed, and commonly known decision support technique. It was originally developed by Emilio Fontela and André Gabus [66] in the early 1970s during the implementation of a research programme by the Batelle Research Institute in Geneva, Switzerland. The project was devoted to the identification of the key general problems of the contemporary world and their perception [67]. The technique provided universal tools for the identification of cause–effect chains. Unfortunately, it was only occasionally applied by others until the turn of the century [68]. Over the past fifteen years, however, it has become increasingly popular [69,70], with a considerable increase in applications in diverse research and practical decision-making areas. The technique underwent many modifications during the course of its successful application to diverse areas and problems [71]. Today, it is commonly used both as a standalone tool [72,73] and to enhance other decision support tools [74–76].

The DEMATEL technique has also been applied to solve transportation- and infrastructure-related engineering problems. Chen and Lee [77] utilised the technique to explore a strategy to develop Taiwan into a tourist transport centre. The identification of key factors was applied with this regard. Dimic et al. [78] analysed the important role of the contemporary transportation system in preserving the environment and sustainable development. The analysis allowed for the identification of a justified sustainable transport strategy. The attitude–behaviour gap in the adoption of a sustainable transportation means was examined by Haider et al. [79]. The technique was applied to identify the key relations between barriers to electric vehicle technology adoption and consumer characteristics. Conversely, Rajak and Dhanalakshmi [80] used the technique during an analysis of the barriers to sustainable transportation. The technique was also successfully applied for specific tasks of the identification of adequate criteria for logistics provider selection (Govindan et al., 2023) [81], an analysis of complex interactions between success factors (Gupta et al., 2023) [82], and the identification of particular behaviours in enterprises (Li et al., 2023) [83].

DEMATEL has also often been used to support other approaches, MCDA tools in particular, while dealing with transportation and infrastructural problems. For example, Liu et al. [84] utilised the technique to support both the DEMATEL-Based Analytic Network Process (DANP) and VIKOR techniques while trying to improve a metro–airport connection service. Ou et al. [85] used DEMATEL to enhance Interpretative Structural Modelling (ISM) while identifying factors that affect rail tunnel construction projects. DEMATEL was also applied to support ANP used to rank factors that hinder the public transportation system in the post-COVID-19 era [86]. Broniewicz and Ogrodnik [87] utilised the technique to provide a necessary means for weighting assessment criteria and to rank six expressway pass alternatives.

It is worth noting that the context of bike transportation also appears in several publications. For example, Ma et al. [88] complemented the VIKOR technique with the application of DEMATEL while assessing the quality of bike-sharing services. And the same techniques were used to determine sustainable development strategies and adoption paths for public bike-sharing services from the perspective of their users [89].

The short literature review presented here proves that DEMATEL comprises a comprehensive and useful tool for dealing with diverse transportation and infrastructural problems. This is mainly because of its original capabilities of analysing the interrelations between influencing factors and providing comprehensive support for other tools.

### 2.3. Gap Analysis and Research Highlights

As shown in the literature review above, many studies focus on planning and policy efforts to increase the use of bicycle transportation. In many cases, initiatives are motivated by a desire to reduce car use and its accompanying environmental consequences, or by pointing out the positive impact on increasing physical activity. Various solutions have been presented in the literature to support the design of safe bicycle infrastructure. Due to the location of bicycle paths in cities, often adjacent to roads used by cars, the problems of polluted air and noise have also been the subject of research. However, the research presented here did not focus on determining the nature and importance of factors influencing the development of bicycle infrastructure.

The DEMATEL method has already been used in the transport and infrastructure sectors. To the best of our knowledge, there has been no research report on its application in identifying and prioritising factors affecting the process of the development of cycling infrastructure.

Poland is a good area for research because it is at an early stage of creating proper and safe systems for cycle paths in cities.

To fill this gap, our study proposes an assessment of the role and importance of factors influencing the development of cycling infrastructure. This is important, and, as demonstrated by the analysis of the publications presented above, there are no studies that have focused on assessing the importance of key factors. This prompted the authors to address this specific research problem. The proposed methodology is a universal one and may also be applied to the assessment of other factors.

## 3. Materials and Methods

### 3.1. Identification of Factors Influencing the Development of Bicycle Paths

In order to identify the factors influencing the development of bicycle paths, the Scopus, Web of Science, and Polish BAZTECH databases were reviewed by searching for publications using the following keyword: bicycle paths. Due to the very broad subject matter of the retrieved articles, the search was limited to the following subject areas: engineering, environmental science, decision sciences, business, management and accounting, and multidisciplinary areas. Figure 2 illustrates how we selected publications containing information on factors that have a key impact on the development of bicycle infrastructure.

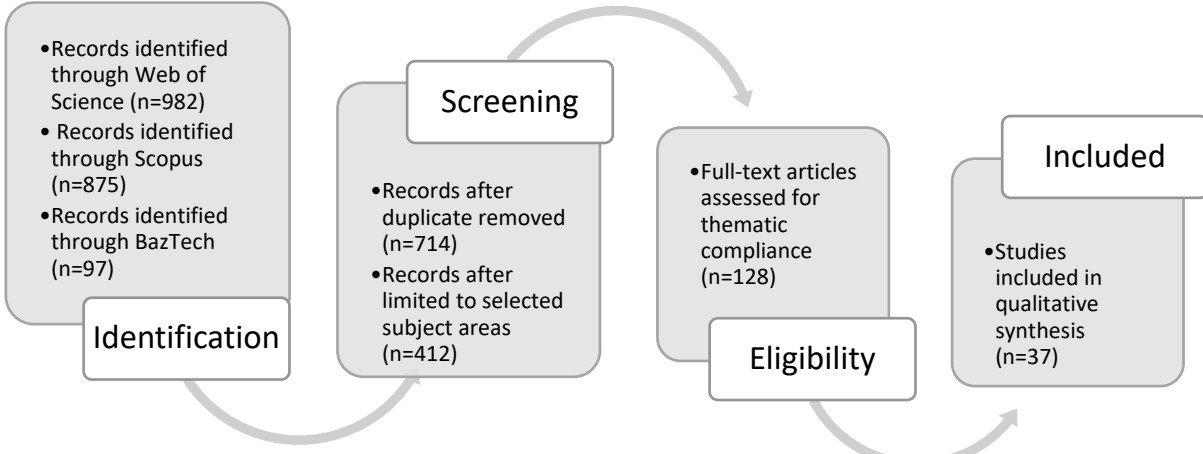

**Figure 2.** The method for selecting publications containing information on the development of bicycle paths.

The table below (Table 1) presents the identified factors with justification and references to the sources.

**Table 1.** Identified factors influencing the development of bicycle infrastructure.

| Factor | Reasons | References |
|---|---|---|
| Planning of bicycle paths, taking into account the separation of individual paths and their continuity, coherence, and length. | Two of the indicators determining the city's readiness for bicycle traffic is the length of paths and the separation of individual bicycle paths. The development of longer bicycle paths and the construction of new bicycle paths result in an increase in the use of bicycles in urban transport. In Polish cities, a major problem is the lack of coherence of the bicycle network. Well-planned cycle paths can reduce traffic congestion. | Zeyed 2016 [21] Roman, Roman 2014 [60] Rybarczyk, Wu 2010 [29] Jarosz, Springer 2021 [1] Bao et al., 2017 [33] |
| Designing bicycle paths, taking into account the safety, space management, terrain, and attractiveness of the surroundings. | Properly planned cycle paths can reduce safety risks for both cyclists and motor vehicle drivers. When designing cycle paths, land use, environmental requirements, socio-economic conditions, and road traffic conditions must be taken into account. Designing safe infrastructure for all categories of travellers, including cyclists, is a fundamental requirement. | Bao et al., 2017 [33] Silvano et al., 2016 [37] Stipancic et al., 2016 [38] Apasnore et al., 2017 [39] Pokorny et al., 2018 [40] Pokorny et al., 2017 [41] Cui et al., 2014 [24] Dondi et al., 2011 [25] Lavrenec et al., 2022 [32] Ribeiro et al., 2022 [14] Vansteenkiste et al., 2017 [48] Xu et al., 2016 [49] |
| Developed infrastructure in the form of bicycle parking lots and parking spaces. | The ability to park bikes results in higher levels of cycling to work. Investment in bicycle parking at railway stations can increase the combined use of bicycles and trains. Cooperation with other urban means of transport improves transport efficiency. | Buehler 2012 [23] Radzimski, Beim 2012 [36] Mamrayeva, Tashenova 2017 [65] Martens 2007 [64] Yang et al., 2018 [61] McBain, Caulfield 2017 [62] |
| Elimination of difficulties (thresholds, faults, surface damage, lack of lighting, etc.). | Bicycle transport infrastructure requires continuous expenditure, both for its modernisation and the construction of new facilities. The technical condition of bicycle paths and the width of bicycle paths influence the behaviour of their users; it is necessary to reduce the speed associated with various types of disturbances. A recommended solution in other cities is the possibility of submitting demands to the city authorities regarding road surface damage and improvement of comfort and safety. | Radzimski, Beim 2012 [36] Bernardi, Rupi 2015 [90] Vansteenkiste et al., 2017 [48] Vansteenkiste et al., 2014 [47] Xu et al., 2016 [49] Roman 2017 [11] |
| Information campaigns encouraging society to cycle for health and environmental protection. | One of the most important barriers hindering the implementation of cycling policy is the negative attitude of society towards cycling; often, convincing people to change their means of transport requires a long-term promotional programme. Young people need to be convinced that this is a convenient, fast, and cheap way of getting around, which may contribute to changing their lifestyle to a more active one. The intensive and continuous promotion of bicycle transport is an important activity that has had the desired effect in the existing and developing trend of bicycle culture in Poland. | Stępień-Słodkowska et al., 2017 [10] Liszka 2013 [2] Eryiğit, Ter 2014 [19] Dzieniowska, Dolińska 2017 [20] |

**Table 1.** *Cont.*

| Factor | Reasons | References |
|---|---|---|
| Promoting cycling by employers (e.g., showers for cyclists). | An employer's encouragement to use a bicycle may be one of the factors influencing more frequent commuting to work by bicycle. The roles of bicycle parking, bicycle showers, free parking spaces, and transport benefits as determinants of cycling to work should be emphasized. | Buehler 2012 [23] Zeyed 2016 [21] |
| Sharing bicycle systems. | The introduction of public bicycle systems (PBSSs) is an element of an effective transport policy. The launch of bicycle rental systems increases interest in their use. The economic benefits provided by bike-sharing systems are related to a reduction in travel time. | Castillo-Manzano et al., 2015 [54] Roman, Roman 2014 [60] Basu, Vasudevan 2013 [22] Bullock et al., 2017 [59] |
| Regulations promoting cycling in the field of transport policy (e.g., preferential treatment of bicycle users at intersections, the possibility of transporting bicycles on public transport). | The solutions implemented in urban transport systems in Western Europe in the field of bicycle policy are an illustration of pro-ecological trends. The development of bicycle paths and public transport should be combined with restrictions on car traffic. There is a need to develop legal regulations supporting the bicycle transport system. | Liszka 2013 [2] Jarosz, Springer 2021 [1] Mamrayeva, Tashenova 2017 [65] |

### 3.2. The Methodology for Determining Key Factors

There are different approaches available that can be applied to support the process of key factor identification. However, the complexity of the contemporary world and gaps in the available yet largely imperfect information make the analysis of real-life problems susceptible to dealing with factors of both a tangible and intangible nature. The application of intangibility-aware techniques for the reliable identification of key factors therefore seems indispensable in such a case. There nevertheless seem to be only a few suitable and established techniques that are aware of not only the tangible but also the intangible nature of factors. For example, techniques like Godet's MICMAC [91] and Interpretative Structural Modelling (ISM) [92]. Nevertheless, both techniques are clearly inferior to the more sophisticated DEMATEL technique. This is mainly because the former techniques provide a very limited capability of comprehensively identifying the actual roles played by the considered factors. However, the explicit processing of information about both the intermediate and total influence of factors via the technique results in a comprehensive exploration of the multi-dimensional role and classification of factors.

DEMATEL's inventors made it capable of dealing with available imperfect and qualitative information. The application of an ordinal scale with crisp number levels for the assessment of direct influence makes this possible. However, many years of DEMATEL's persistence and development have resulted in diverse proposals for expressing ordinal scale levels using non-crisp numbers. For example, several concepts like fuzzy and grey numbers have been applied in this regard [71]. One must nevertheless be aware that there are plenty of such non-standard information representations available. Their application usually results in a lot more complex calculations and other inconveniences, e.g., a real risk of distortion in the processed information due to the use of arbitrarily chosen techniques to make the non-crisp analysis results meaningful. Moreover, the actual merits of the replacement of crisp assessments of direct influence with non-crisp assessments also seem arguable [93]. This is why the legacy of crisp DEMATEL is consciously utilised in this paper.

### 3.3. DEMATEL Technique

Crisp data and the calculation mechanisms of the original DEMATEL [67] are applied in this paper to identify the key factors in the planning and design of bicycle paths. The principal concepts applied by DEMATEL in the case of the considered factors are presented in Table 2 (inputs) and Table 3 (results).

**Table 2.** Principal input concepts of DEMATEL methodology (inputs).

| Concept | Description |
|---|---|
| Direct influence | Direct action of a contextual nature, which is exercised by the i-th consecutive entity on the consecutive j-th entity, out of n considered entities ($i, j = 1, 2 \ldots$ n) |
| Direct influence scale | The ordinal scale of direct influence 0-N with the following fixed steps: 0 (no direct influence at all), N (extreme direct influence), where N denotes arbitrary positive integer (originally: N = 4); the intermediate scale levels from 1 to N − 1 express gradual increase in direct influence |
| Direct influence intensity $x_{ij}^*$ | Measure of the intensity of direct influence of the i-th consecutive entity on the j-th consecutive entity, out of n considered entities ($i, j = 1, 2 \ldots$ n) <br> In the case of the engagement of K experts, direct influence intensity for the k-th consecutive expert is denoted by $x_{ij}^{*(k)}$ |
| Structure of direct influence | Full set of direct influences between n considered entities |
| Matrix of direct influence X* | Raw matrix of direct influence: <br> $\mathbf{X}^* = \left[ x_{ij}^* \right]_{n \times n}$ <br> In the case of the engagement of K experts, each of them provides an individual raw matrix of direct influence, which is denoted by $\mathbf{X}^{*(k)}$ in the case of the k-th consecutive experts, and the raw matrix becomes the average direct influence matrix in the following case: <br> $\mathbf{X} = \frac{\sum_{k=1}^{K} \mathbf{X}^{*(k)}}{K}$ |
| Normalised matrix of direct influence X | A suitable form of a matrix of direct influence: <br> $\mathbf{X}$: $\lim_{k \to \infty} \mathbf{X}^k = \mathbf{0}_{n \times n}$ |
| Graph of direct influence G($\mathbf{X}^*$) | The alternative representation of the structure of direct influence using a digraph |

**Table 3.** Principal input concepts of DEMATEL methodology (results).

| Concept | Description |
|---|---|
| Indirect influence | Action of a contextual nature, which is exercised by the i-th consecutive entity on the j-th consecutive entity, out of n considered entities, through another considered entity (or even more entities) ($i, j = 1, 2 \ldots$ n) |
| Indirect influence intensity $\Delta x_{ij}$ | Measure of the intensity of indirect influence of the i-th consecutive entity on the j-th consecutive entity, out of n considered entities <br> ($i, j = 1, 2 \ldots$ n) |
| Structure of indirect influence | Full set of indirect influences between n considered entities |
| Matrix of indirect influence ΔX | Concise mathematical representation of the structure of indirect influence: <br> $\Delta \mathbf{X} = \left[ \Delta x_{ij} \right]_{n \times n} = \mathbf{X}^2 (\mathbf{I} - \mathbf{X})^{-1}$ |
| Graph of indirect influence G(ΔX) | Alternative representation of indirect influence structure employing a digraph |
| Total influence | The combination of direct and indirect influences, which are exercised by the i-th consecutive entity on the j-th consecutive entity, out of n considered entities |

<div align="center">**Table 3.** *Cont.*</div>

| Concept | Description |
|---|---|
| Structure of total influence | Full set of total influences between n considered entities |
| Total influence intensity $t_{ij}$ | Measure of the intensity of total influence of the i-th consecutive entity on the j-th consecutive entity, out of n considered entities ($i, j = 1, 2 \ldots$ n) |
| Matrix of total influence **T** | Concise mathematical representation of the structure of indirect influence: $\mathbf{T} = \left[\Delta t_{ij}\right]_{n \times n} = \mathbf{X} + \Delta \mathbf{X} = \mathbf{X}\left(\mathbf{I} - \mathbf{X}\right)^{-1}$ |
| Graph of total influence G(**T**) | Alternative representation of total influence structure utilising a digraph application |
| Net total influence structure | Structure of resulting (net) total influence |
| Net total influence intensity $t_{\mathrm{nt}_{ij}}$ | The measure of the resulting intensity of total influence of the i-th consecutive entity on the j-th consecutive entity, out of n considered entities ($i, j = 1, 2 \ldots$ n): $t_{\mathrm{nt}_{ij}} = t_{\mathrm{nt}_{ij}} - t_{\mathrm{nt}_{ji}} > 0$ |
| Net total influence matrix $\mathbf{T}_{\mathrm{nt}}$ | Concise mathematical representation of the structure of indirect influence: $\mathbf{T}_{\mathrm{nt}} = \left[t_{\mathrm{nt}_{ij}}\right]_{n \times n}$ |
| Net total influence digraph G($\mathbf{T}_{\mathrm{nt}}$) | Alternative representation of the net total influence structure through a digraph |
| Prominence s+ | Interrelation between the i-th consecutive entity and other entities, out of n considered entities ($i = 1, 2 \ldots$ n): $s_i^+ = \sum_j^i t_{ij} + t_{ji}$ |
| Relation s− | Causality of the i-th consecutive entity ($i = 1, 2 \ldots$ n): $s_i^+ = \sum_j^i t_{ij} - t_{ji}$ |
| Cause–effect diagram | $s^+$ vs. $s^-$ diagram for the classification of n considered entities |

The DEMATEL technique is finally applied in the following manner in this paper:

1. Individual questionnaire survey results are aggregated independently to obtain the average direct influence matrix **X**\* in the case of complete survey results (**X**e\* in the case of results based on the opinions of the invited experts only) to define the direct influence structure of the factors.

2. The average influence **X** matrix is derived for the case of complete survey results according to Table 2 (**X**e in the case of the application of the expert-related results only).

3. The total influence matrix T is obtained according to Table 3 (**T**e in the case of the results provided by the opinions of the experts only) to obtain the structure of the total influence of the considered factors.

4. Prominence and relation indices are calculated for each factor according to Table 3 to develop cause–effect diagrams for the factors.

5. The net total influence matrix **T**nt is derived in the case of complete survey results according to Table 3 (**T**nt e in the case of the application of the expert-related results only).

### 3.4. Input Data

Eight factors (n = 8) that influence the process of the development of bicycle paths were selected based on the literature. The factors were coded using symbols A1 to A8. The meaning of the factors is presented in Table 4.

A 56-question questionnaire was then prepared, in which respondents were asked to rate, on a five-point scale, the impact of each of the eight factors on each other. A group of five invited professional experts (*K* = 5) were asked to complete the questionnaire. The experts represented the following fields and specialisations:

- An administration employee involved in the planning of bicycle paths;
- An administration employee dealing with the development of bicycle paths;
- An administration employee involved in the planning and designing of transport systems in prepared planning documents;

- An administration employee involved in the monitoring of bicycle paths;
- An administration employee involved in the strategic management of city development.

A group of eighteen bicycle users ($K = 18$) were asked to complete the questionnaire. The users represented the following ways of using a bicycle:

- For commuting to work most days of the year;
- For commuting to work only during the summer months;
- For recreational purposes.

**Table 4.** The meaning of factors and their denotations.

| Factor | Meaning |
|---|---|
| A1 | Planning of bicycle paths, taking into account the separation of individual paths and their continuity, coherence, and length |
| A2 | Design of bicycle paths, taking into account the safety, space management, terrain, and attractiveness of the surroundings |
| A3 | The development of infrastructure in the form of bicycle parking lots and parking spaces |
| A4 | The elimination of obstacles (thresholds, faults, surface damage, path lighting, etc.) |
| A5 | Information campaigns encouraging society to cycle for health and environmental protection |
| A6 | Promotion of cycling by employers (e.g., through showers for cyclists) |
| A7 | Introduction of bicycle-sharing systems |
| A8 | Regulations promoting cycling in the field of transport policy (e.g., preferential treatment of bicycle users at intersections, the possibility of transporting bicycles on public transport) |

The questionnaire survey provided two data sets for DEMATEL. The first data set contained complete and valid data from all twenty-three survey participants (see Table 5), while the second one was limited to the opinions provided by the five invited experts with professional backgrounds in the decision making of bicycle path planning and design (see Table 6).

**Table 5.** Average direct influence matrix **X\*** components for all questionnaire survey participants.

|  | A1 | A2 | A3 | A4 | A5 | A6 | A7 | A8 |
|---|---|---|---|---|---|---|---|---|
| **A1** | 0 | 3.4444 | 3.1667 | 2.8333 | 2.7778 | 2.1111 | 2.6111 | 2.7778 |
| **A2** | 3.2778 | 0 | 3.1667 | 2.8889 | 2.5 | 2.2222 | 2.7778 | 3.0556 |
| **A3** | 2.7778 | 2.5556 | 0 | 2.4444 | 2.5 | 2.2222 | 2.5 | 2.5556 |
| **A4** | 2.9444 | 3 | 2.7778 | 0 | 2.3889 | 1.8889 | 2.1111 | 2.3889 |
| **A5** | 2.7222 | 2.0556 | 2.3333 | 2 | 0 | 2.1667 | 2.3333 | 2.5556 |
| **A6** | 1.8889 | 1.6667 | 1.8333 | 1.5556 | 2.2778 | 0 | 1.8333 | 1.9444 |
| **A7** | 2 | 2.1111 | 2.2778 | 1.6111 | 2.5 | 1.8889 | 0 | 1.8889 |
| **A8** | 2.6667 | 2.9444 | 2.7222 | 2.5 | 2.7778 | 2.2778 | 2.7222 | 0 |

The corresponding graphs of direct influence for the overall questionnaire survey results and expert opinion-based results are presented in Figure 3 and in Figure 4, respectively. Note that the arc line styles applied in the figures correspond to the following classes due to direct influence intensity exceeding the following direct influence scale level:

- 3 (bold line style);
- 2 (dashed line style);
- 1 (dotted line style).

**Table 6.** Average direct influence matrix $\mathbf{X}_e^*$ components for expert opinions.

|     | A1  | A2  | A3  | A4  | A5  | A6  | A7  | A8  |
| --- | --- | --- | --- | --- | --- | --- | --- | --- |
| **A1** | 0   | 3.8 | 3.5 | 2.4 | 2.8 | 2   | 2   | 3.6 |
| **A2** | 3.2 | 0   | 3.4 | 2.6 | 2.4 | 1.8 | 2.4 | 3.4 |
| **A3** | 2.4 | 2.2 | 0   | 1.6 | 2   | 2.2 | 2.4 | 2.4 |
| **A4** | 2.6 | 2.8 | 2   | 0   | 2.6 | 2   | 1.8 | 2.8 |
| **A5** | 3   | 2.6 | 2.2 | 1.6 | 0   | 2.4 | 2.8 | 3   |
| **A6** | 1.4 | 1.8 | 2   | 1.6 | 2.6 | 0   | 1.6 | 2.4 |
| **A7** | 1.6 | 1.8 | 1.8 | 1.4 | 3.2 | 1.6 | 0   | 2.2 |
| **A8** | 3.4 | 3.4 | 3.2 | 3.2 | 3.4 | 2   | 3   | 0   |

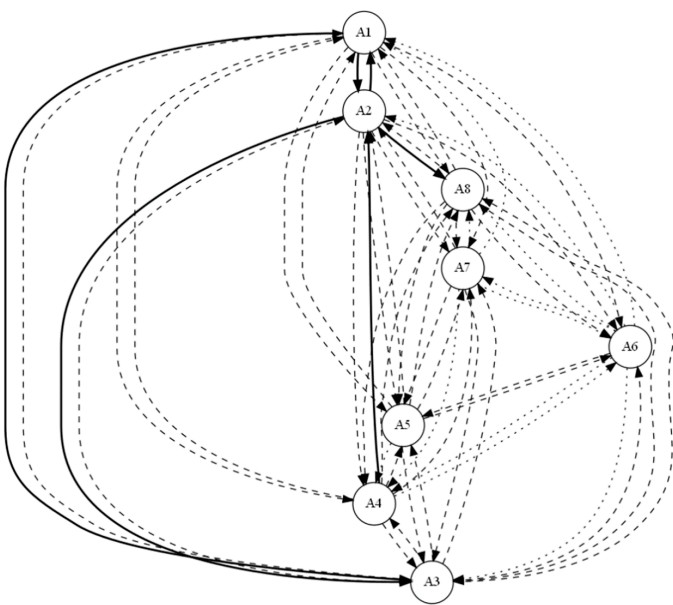

**Figure 3.** Graph of direct influence (overall survey results).

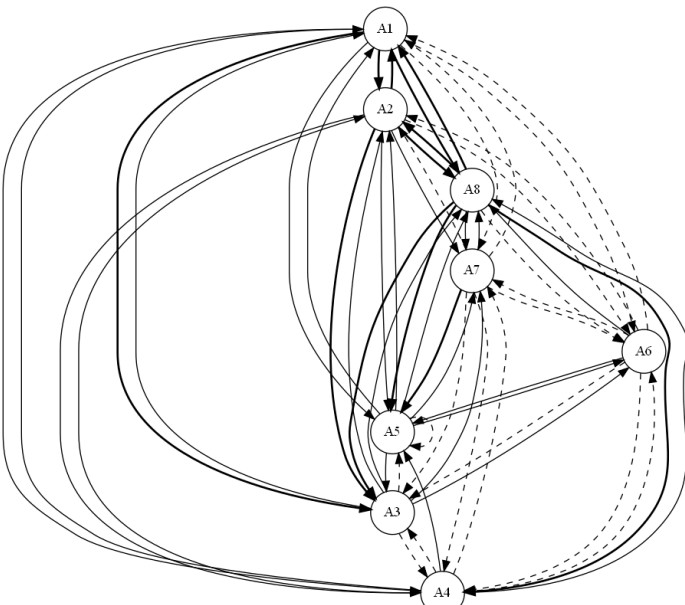

**Figure 4.** Graph of direct influence (expert opinion-based questionnaire survey results only).

## 4. Results

This section is devoted to the presentation of the results of the application of the DEMATEL technique for the identification of the key factors in the process of the design of bicycle path networks.

### 4.1. Overall Questionnaire Survey

The total influence matrix **T** obtained for the overall questionnaire survey results is presented in Table 7.

**Table 7.** Total influence matrix **T** components for overall questionnaire survey results.

| Factor | A1 | A2 | A3 | A4 | A5 | A6 | A7 | A8 |
|--------|------|------|------|------|------|------|------|------|
| **A1** | 0.8264 | 0.9551 | 0.9458 | 0.8542 | 0.9232 | 0.7757 | 0.8863 | 0.9046 |
| **A2** | 0.9722 | 0.8121 | 0.9503 | 0.8603 | 0.9173 | 0.7839 | 0.8972 | 0.9193 |
| **A3** | 0.8624 | 0.8368 | 0.7239 | 0.7626 | 0.8298 | 0.7099 | 0.8020 | 0.8143 |
| **A4** | 0.8563 | 0.8419 | 0.8144 | 0.6423 | 0.8119 | 0.6850 | 0.7740 | 0.7957 |
| **A5** | 0.8059 | 0.7643 | 0.7763 | 0.6970 | 0.6662 | 0.6635 | 0.7452 | 0.7632 |
| **A6** | 0.6462 | 0.6239 | 0.6318 | 0.5666 | 0.6462 | 0.4616 | 0.6068 | 0.6189 |
| **A7** | 0.7029 | 0.6922 | 0.7008 | 0.6149 | 0.7052 | 0.5909 | 0.5707 | 0.6660 |
| **A8** | 0.8972 | 0.8896 | 0.8825 | 0.7991 | 0.8784 | 0.7443 | 0.8473 | 0.7375 |

Table 8 contains information about the prominence and relation indices, and Figure 5 presents a cause–effect diagram for the overall questionnaire survey results. The identified roles of consecutive factors according to the prominence and the relation indices are presented in Table 9.

**Table 8.** The values of prominence and relation indices for overall questionnaire survey results.

| Factor | Prominence | Relation |
|--------|------------|----------|
| **A1** | 13.6408 | 0.501816 |
| **A2** | 13.5285 | 0.696634 |
| **A3** | 12.7675 | −0.084283 |
| **A4** | 12.0184 | 0.424825 |
| **A5** | 12.2597 | −0.496765 |
| **A6** | 10.2169 | −0.612946 |
| **A7** | 11.3729 | −0.885826 |
| **A8** | 12.8953 | 0.456545 |

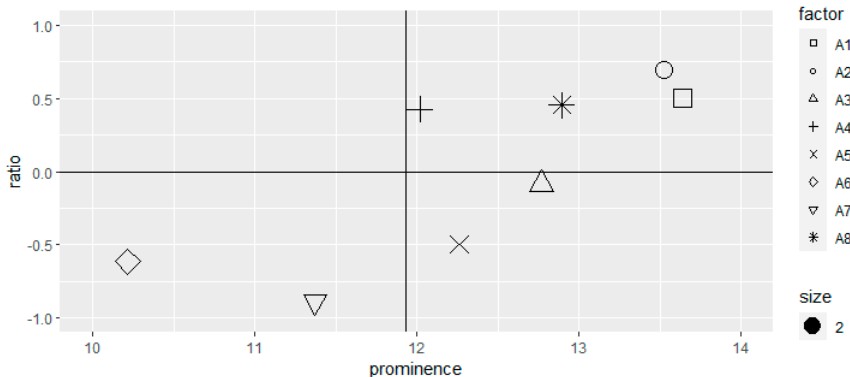

**Figure 5.** Cause–effect diagram (overall questionnaire survey results).

**Table 9.** Individual rankings and roles of factors (overall questionnaire results).

| Factor | Prominence Rank | Relation Rank |
|---|---|---|
| **A1** | 1 (Very High) | 2 (Cause) |
| **A2** | 2 (Very High) | 1 (Cause) |
| **A3** | 4 (High) | 5 (Neutral) |
| **A4** | 6 (Average) | 4 (Cause) |
| **A5** | 5 (Average) | 6 (Effect) |
| **A6** | 8 (Very Low) | 7 (Effect) |
| **A7** | 7 (Low) | 8 (Effect) |
| **A8** | 3 (High) | 3 (Cause) |

And, finally, the net total influence matrix $\mathbf{T}_{nt}$ components and a graph of the net total influence are presented in Table 10 and Figure 6, respectively.

**Table 10.** Net total influence matrix $\mathbf{T}_{nt}$ components for overall questionnaire survey results.

| Factor | A1 | A2 | A3 | A4 | A5 | A6 | A7 | A8 |
|---|---|---|---|---|---|---|---|---|
| **A1** | 0 | 0 | 0.0834 | 0 | 0.1173 | 0.1295 | 0.1835 | 0.0073 |
| **A2** | 0.0171 | 0 | 0.1135 | 0.0183 | 0.1530 | 0.1601 | 0.2050 | 0.0297 |
| **A3** | 0 | 0 | 0 | 0 | 0.0535 | 0.0780 | 0.1012 | 0 |
| **A4** | 0.0022 | 0 | 0.0519 | 0 | 0.1150 | 0.1184 | 0.1591 | 0 |
| **A5** | 0 | 0 | 0 | 0 | 0 | 0.0173 | 0.0400 | 0 |
| **A6** | 0 | 0 | 0 | 0 | 0 | 0 | 0.0159 | 0 |
| **A7** | 0 | 0 | 0 | 0 | 0 | 0 | 0 | 0 |
| **A8** | 0 | 0 | 0.0682 | 0 | 0.1153 | 0.1254 | 0.1813 | 0 |

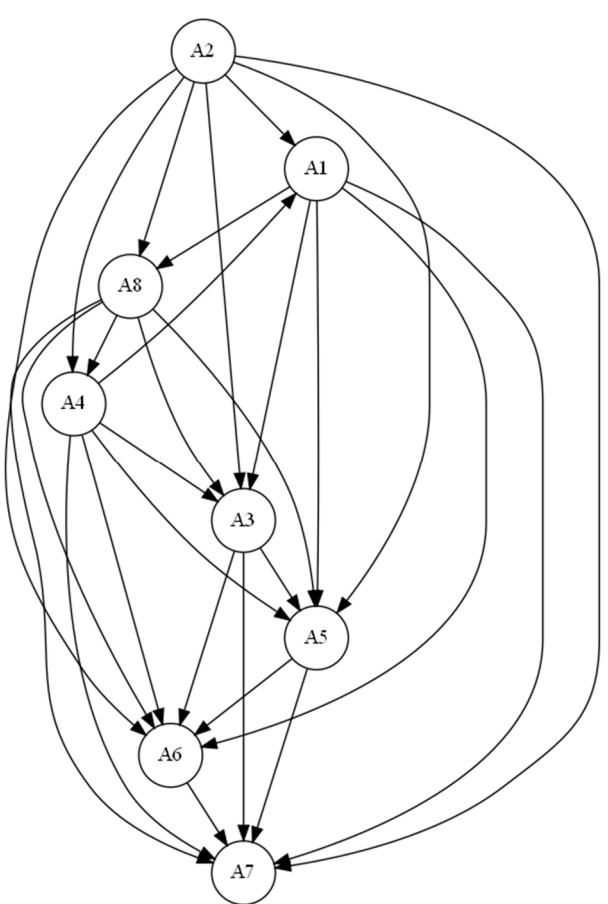

**Figure 6.** The net total influence graph (overall questionnaire survey results).

### 4.2. Expert Opinion-Based Questionnaire Survey Results

The total influence matrix **T** obtained for the expert opinion-based questionnaire survey results is presented in Table 11.

**Table 11.** Total influence matrix **T** components for expert opinions only.

| Factors | A1 | A2 | A3 | A4 | A5 | A6 | A7 | A8 |
|---|---|---|---|---|---|---|---|---|
| **A1** | 0.5058 | 0.6720 | 0.6569 | 0.5190 | 0.6422 | 0.4866 | 0.5463 | 0.6959 |
| **A2** | 0.6113 | 0.4982 | 0.6295 | 0.5072 | 0.6050 | 0.4612 | 0.5400 | 0.6637 |
| **A3** | 0.4833 | 0.4891 | 0.3929 | 0.3880 | 0.4897 | 0.3994 | 0.4516 | 0.5210 |
| **A4** | 0.5322 | 0.5534 | 0.5186 | 0.3524 | 0.5523 | 0.4229 | 0.4645 | 0.5797 |
| **A5** | 0.5635 | 0.5642 | 0.5449 | 0.4365 | 0.4657 | 0.4526 | 0.5195 | 0.6066 |
| **A6** | 0.4058 | 0.4311 | 0.4347 | 0.3537 | 0.4686 | 0.2751 | 0.3845 | 0.4756 |
| **A7** | 0.4188 | 0.4366 | 0.4321 | 0.3496 | 0.4971 | 0.3485 | 0.3205 | 0.4741 |
| **A8** | 0.6681 | 0.6854 | 0.6717 | 0.5694 | 0.6938 | 0.5087 | 0.6075 | 0.5825 |

Table 12 shows the values obtained for the prominence and relation indices for the questionnaire survey results obtained for the expert opinions only. The corresponding cause–effect diagram is presented in Figure 7. The ranks and identified roles of consecutive factors according to the prominence and the relation indices, respectively, are presented in Table 13.

**Table 12.** The values of prominence and relation indices for expert opinions only.

| Factor | Prominence | Relation |
|---|---|---|
| **A1** | 8.913566 | 0.5357729 |
| **A2** | 8.846053 | 0.1863695 |
| **A3** | 7.896194 | −0.6663403 |
| **A4** | 7.451752 | 0.5001212 |
| **A5** | 8.568022 | −0.2607674 |
| **A6** | 6.584140 | −0.1260405 |
| **A7** | 7.111957 | −0. 5571755 |
| **A8** | 9.586232 | 0.3880602 |

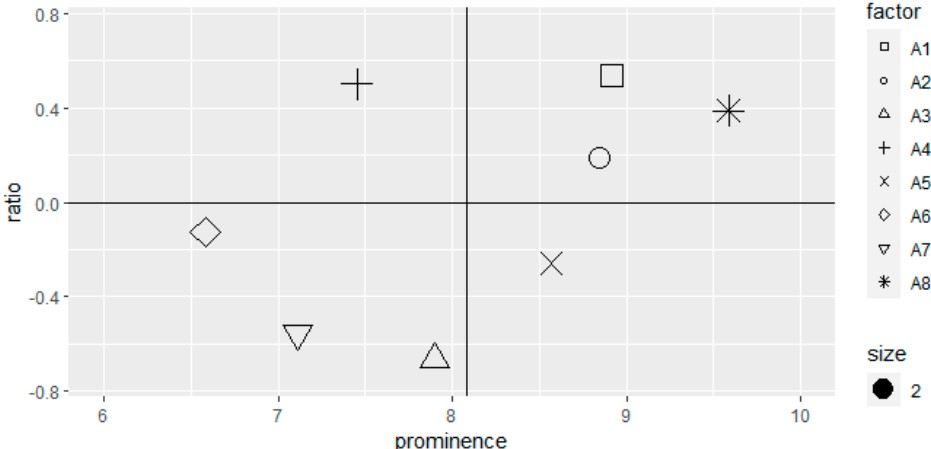

**Figure 7.** Cause–effect diagram (expert-based questionnaire survey results only).

**Table 13.** Individual ranking of factors (expert-based questionnaire results only).

| Factor | Prominence Rank | Relation Rank |
|--------|-----------------|---------------|
| **A1** | 2 (High) | 1 (Cause) |
| **A2** | 3 (High) | 4 (Cause) |
| **A3** | 5 (Average) | 6 (Effect) |
| **A4** | 6 (Low) | 2 (Cause) |
| **A5** | 4 (Average) | 8 (Effect) |
| **A6** | 8 (Very Low) | 5 (Effect) |
| **A7** | 7 (Very Low) | 7 (Effect) |
| **A8** | 1 (Very High) | 3 (Cause) |

The net total influence matrix $\mathbf{T}_{nt\,e}$ components and the corresponding graph of the net total influence are finally presented in Table 14 and Figure 8, respectively.

**Table 14.** Net total influence matrix $\mathbf{T}_{nt}$ components for expert opinion-based questionnaire survey results only.

| Factor | A1 | A2 | A3 | A4 | A5 | A6 | A7 | A8 |
|--------|------|------|------|------|------|------|------|------|
| **A1** | 0 | 0.0607 | 0.1736 | 0 | 0.0787 | 0.0808 | 0.1274 | 0.0278 |
| **A2** | 0 | 0 | 0.1404 | 0 | 0.0409 | 0.0301 | 0.1035 | 0 |
| **A3** | 0 | 0 | 0 | 0 | 0 | 0 | 0.0195 | 0 |
| **A4** | 0.0132 | 0.0462 | 0.1305 | 0 | 0.1157 | 0.0692 | 0.1149 | 0.0104 |
| **A5** | 0 | 0 | 0.0552 | 0 | 0 | 0 | 0.0225 | 0 |
| **A6** | 0 | 0 | 0.0353 | 0 | 0.0160 | 0 | 0.0360 | 0 |
| **A7** | 0 | 0 | 0 | 0 | 0 | 0 | 0 | 0 |
| **A8** | 0 | 0.0217 | 0.1507 | 0 | 0.0872 | 0.0332 | 0.1334 | 0 |

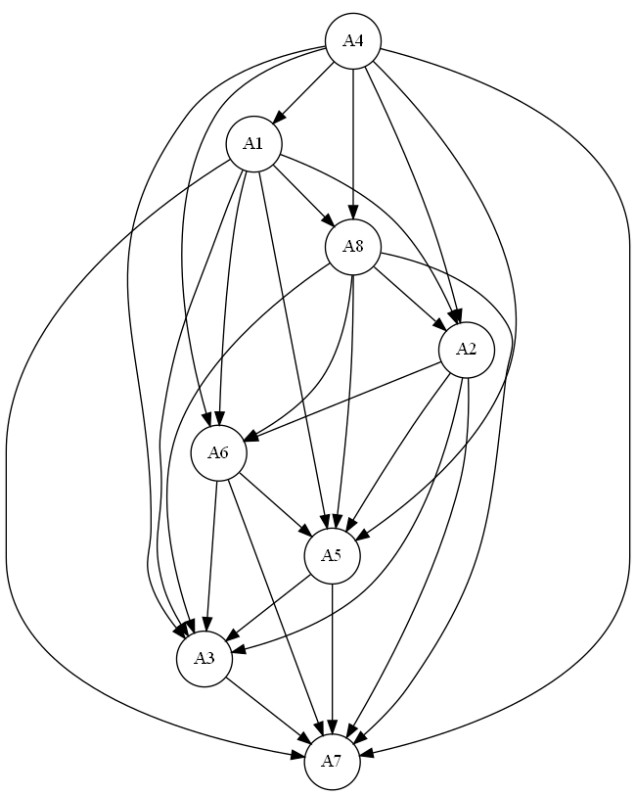

**Figure 8.** The net total influence graph (expert opinions only, questionnaire survey results).

## 5. Discussion

Let us start the discussion with the overall results of the questionnaire survey, which are presented in Section 3. It is clear from the relation index values presented in Table 9 that, among the considered factors, there are four factors that play key causal roles concerning bicycle path network development, namely, the planning of bicycle paths, taking into account the separation of individual paths and their continuity, coherence, and length (A1); regulations promoting cycling in the field of transport policy (A8); the elimination of obstacles (A4); and the design of bicycle paths, taking into account the safety, space management, terrain, and attractiveness of the surroundings (A2). However, several factors are clear effects of the influence of other factors (particularly the promotion of cycling by employers—A6, information campaigns encouraging society to cycle for health and environmental protection—A5, and the introduction of bicycle-sharing systems—A7). The value of the relation index obtained for the remaining factor of the development of infrastructure in the form of bicycle parking lots and parking spaces (A3) confirms its rather neutral (neither causal nor effective) role.

The contents of Table 9 also confirm that the introduction of bicycle-sharing systems and the promotion of cycling by employers are the only entities that are rather weakly connected with the other entities. This fact further strengthens the primary role of all identified key factors in the process of the total influence distribution among all entities. Note that the importance of the effect called information campaigns encouraging society to cycle for health and environmental protection amongst all identified effects is also strengthened for the same reason.

The above conclusions are also confirmed by the classification of the considered factors according to the cause–effect diagram in Figure 5. This is because occupying the right upper quarter of the diagram are all the causes: the planning of bicycle paths, taking into account the separation of individual paths and their continuity, coherence, and length; regulations promoting cycling in the field of transport policy; the elimination of obstacles; and the design of bicycle paths, taking into account the safety, space management, terrain, and attractiveness of the surroundings. Such a location corresponds well with both the causality of the factors and their strong linkage to other factors concerning the total influence. However, the factor of information campaigns encouraging society to cycle for health and environmental protection is located in the right lower quarter, which corresponds to a strong connection with the other factors. And, finally, the significance of the factor of the promotion of cycling by employers and the factor of the introduction of bicycle-sharing systems seem to suffer a bit from their location in the left lower quarter, which covers the case of a rather weaker connection with the other factors.

All in all, the overall results of the questionnaire survey show that there are four factors:

- The planning of bicycle paths, taking into account the separation of individual paths and their continuity, coherence, and length;
- Regulations promoting cycling in the field of transport policy;
- The elimination of obstacles;
- The design of bicycle paths, taking into account the safety, space management, terrain, and attractiveness of the surroundings in particular

that play key roles in the planning and design of bicycle paths.

And, the other three factors:

- Promotion of cycling by employers;
- Introduction of bicycle-sharing systems;
- Information campaigns encouraging society to cycle for health and environmental protection.

comprise undoubted effects, while the development of infrastructure in the form of bicycle parking lots and parking spaces plays a rather neutral role concerning the transmission of total influence between the considered factors. Note that this observation is compliant with a hierarchical digraph presenting a net total influence structure in Figure 6. This is because

all the causes are located in the upper part of the structure, and all the effects are present in the lower part, with the neutral factor located just in between.

Let us now take a look at the case of the experts' opinions only. The contents of Table 13 show that, according to the experts, the elimination of obstacles (A4) and the planning of bicycle paths, taking into account the separation of individual paths and their continuity, coherence, and length (A1) in particular comprise the most significant causes. Another cause, namely, regulations promoting cycling in the field of transport policy (A8) prevailed over both causes a bit. And the design of bicycle paths, taking into account the safety, space management, terrain, and attractiveness of the surroundings (A2) proves to be the weakest cause. However, the introduction of bicycle-sharing systems (A7) and the development of infrastructure in the form of bicycle parking lots and parking spaces (A3) in particular are the most notable effects. The factors of information campaigns encouraging society to cycle for health and environmental protection (A5) and the promotion of cycling by employers (A6) in particular seem to act as weak effects only. Note that, this time, there is no neutral factor at all.

The contents of Table 13 also confirm that there are four factors (information campaigns encouraging society to cycle for health and environmental protection; the design of bicycle paths, taking into account the safety, space management, terrain, and attractiveness of the surroundings; the planning of bicycle paths, taking into account the separation of individual paths and their continuity, coherence, and length; and regulations promoting cycling in the field of transport policy in particular) that are strongly connected to the other factors and four factors that are rather weakly (the development of infrastructure in the form of bicycle parking lots and parking spaces) or even considerably weakly (the elimination of obstacles, introduction of bicycle-sharing systems, and promotion of cycling by employers in particular) connected to the other factors.

The classification of the factors (see Figure 7) confirms the above conclusions again. This is because the following three factors occupy the right upper quarter in Figure 7, which expresses the causal and strongly connected nature of the factors:

- The design of bicycle paths, taking into account the safety, space management, terrain, and attractiveness of the surroundings.
- Regulations promoting cycling in the field of transport policy.
- The planning of bicycle paths, taking into account the separation of individual paths and their continuity, coherence and length.
- This time, one of the causal factors (the elimination of obstacles) is also located in the upper left quarter, and such a location weakens its significance, a little bit, amongst the identified causes due to a lower connection with the other factors.
- Information campaigns encouraging society to cycle for health and environmental protection profits from the occupation of the lower right quarter in Figure 7 because of its considerably stronger connection with the other factors than the other two remaining effects (introduction of bicycle-sharing systems and promotion of cycling by employers), which are located in the lower left quarter.

All in all, the results of the application of opinions, solely provided by the invited experts, reveal four causes again. The causes include the weakest cause design of bicycle paths, taking into account the safety, space management, terrain, and attractiveness of the surroundings, as well as the following:

- Regulations promoting cycling in the field of transport policy;
- The elimination of obstacles;
- The planning of bicycle paths, taking into account the separation of individual paths and their continuity, coherence, and length.

There is a secondary weak effect, namely, promotion of cycling by employers, a more confident effect, namely, information campaigns encouraging society to cycle for health and environmental protection, as well as the following undoubted effects:

- The introduction of bicycle-sharing systems.

- The development of infrastructure in the form of bicycle parking lots and parking spaces.
- The resulting hierarchical net total influence structure (see Figure 8) acknowledges the above conclusions again. This is because four causes are located in the upper levels of the structure, and the effects occupy its lower levels.

Note that the results of the overall questionnaire survey and the application of only the expert opinions differ. This fact is confirmed by Figure 9, which illustrates similarities and differences in the net total influence between the results obtained for all questionnaire survey results and the results provided by the expert opinions only. Note that the bold digraph arcs in Figure 9 express common net total influence directions. However, the dotted and dashed arcs correspond to the application of all questionnaire survey results and the results based on the expert opinions only, respectively. It seems that the differences between the net total structures deal with almost 40% of individual net total influence relations between the distinct factors. It nevertheless seems that the differences between the two net total influence structures are rather insignificant. That is because they only appear within the factor groups occupying the lower or upper portions of the structures shown in both Figures 6 and 8.

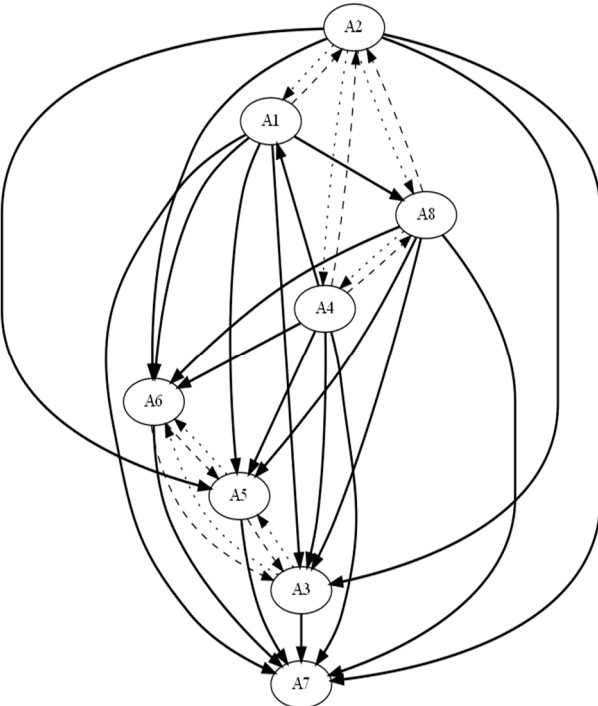

**Figure 9.** The comparison of net total influence structures—the application of complete survey results vs. expert opinions only—see explanations in text. Source: own work.

Let us consider some data to identify the core differences between both sets of results. Table 15 presents a qualitative comparison of the obtained results. Firstly, six distinct classes are used to differentiate the levels of the strength of the connections of factors with the other factors according to the prominence index. The following upper-class limits (relative to the maximum prominence score) are applied as follows: 20% (class denoted by: "—"), 40% ("−"), 50% ("−/+"), 60% ("+/"), 80% ("+"), and 100% ("++"). Secondly, the ranks of the factors in distinct hierarchies of causes (C) and effects (E) according to the ratio index value are presented. The ranks are given between parenthesis signs. Note that letter N expresses the neutrality of a factor.

**Table 15.** Difference in questionnaire survey results.

| Factor | Overall | | Experts | |
|---|---|---|---|---|
| | Link | C/E | Link | C/E |
| **A1** | ++ (1) | C (2) | + (2) | C (1) |
| **A2** | ++ (2) | C (1) | + (3) | C (4) |
| **A3** | + (4) | N (1) | +/− (5) | E (1) |
| **A4** | +/− (6) | C (4) | − (6) | C (2) |
| **A5** | +/− (5) | E (3) | +/− (4) | E (3) |
| **A6** | −− (8) | E (2) | −− (8) | E (4) |
| **A7** | − (7) | E (1) | −− (7) | E (2) |
| **A8** | + (3) | C (3) | ++ (1) | C (3) |

Source: own work.

It is evident in the contents of Table 15 that almost all factors retain their role, regardless of whether the opinions of all respondents or only the opinions provided by the experts are considered. The development of infrastructure in the form of bicycle parking lots and parking spaces (A3) is the only exception in this regard. Its state balances between neutral and effect roles only. However, this is the primary effect role in the case of the data provided by the experts.

Let us now take a look at the causes. It seems that the design of bicycle paths, taking into account the safety, space management, terrain, and attractiveness of the surroundings (A2) and the planning of bicycle paths, taking into account the separation of individual paths and their continuity, coherence, and length (A1) behave more consistently than the other identified constant causes: regulations promoting cycling in the field of transport policy (A8) and the elimination of obstacles (A4). Moreover, the A2 and A1 factors proved to be the most significant causes in both cases of considered data. The A8 and A4 factors behave similarly, but the A8 factor is more consistently interconnected with the other factors. All in all, it seems that the causes should be considered in the order of A2, A1, A8, and A4 when trying to deal with them.

The introduction of bicycle-sharing systems (A7) seems to behave the most consistently among the identified effects. Moreover, it proves to be, alongside the development of infrastructure in the form of bicycle parking lots and parking spaces (A3), the most significant factor for one of the considered questionnaire survey result cases at least. Although the remaining constant effects, namely, information campaigns encouraging society to cycle for health and environmental protection (A5) and the promotion of cycling by employers (A6), behave similarly to each other in both cases, it is the former that stays more interconnected with the other factors. Thus, the following order of the effects is recommended when considering them in detail: A7, A3, A5, and A6.

## 6. Conclusions

The development of bicycle paths is one of the priorities of sustainable city planning. People professionally involved in the development of bicycle paths must take into account many different factors influencing this process. By analysing the literature, it was possible to identify eight factors affecting the development of bicycle infrastructure. Subsequently, the opinions of experts and cyclists specifying the mutual influences of the factors were obtained.

The DEMATEL method was used to achieve the research goal. It allowed for the efficient identification of all cause–effect relationships and key factors, even in the case of complex processes.

The use of knowledge provided by all questionnaire survey participants and the DEMATEL method allowed for the following:

- The identification of the roles of the factors (cause–effect);
- The prioritisation of the factors;

- The indication of differences between the opinions of bicycle users and experts professionally involved in the development of bicycle paths.

The overall outcomes and outcomes based on the opinions of the invited experts show some differences with regard to the order of factors according to the prominence and the relation indices. However, both outcomes are rather consistent with regard to the indication of the actual roles of the considered factors. Thus, the conducted research allowed for the identification of the factors of key importance for the development of bicycle paths. These included the following:

- The designing of bicycle paths, taking into account the safety, space management, terrain, and attractiveness of the surroundings;
- The planning of bicycle paths, taking into account the separation of individual paths and their continuity, consistency, and length;
- The legal regulations promoting cycling;
- The elimination of technical difficulties on bicycle paths.

These factors were indicated as priorities by the users of the bike paths. In contrast, the priority factors perceived by the experts were in the following order:

- The planning of bicycle paths, taking into account the separation of individual paths and their continuity, consistency, and length;
- The elimination of technical difficulties on bicycle paths;
- The legal regulations promoting cycling;
- The design of bicycle paths, taking into account the safety, space management, terrain, and attractiveness of the surroundings.

And the remaining factors proved to be the clear effects under the influence of the causes.

The conducted research is the realisation of an innovative idea of the authors. Previous studies concentrated on the identification of factors influencing the realisation and use of cycling paths. The research contained in the present article focuses on identifying the role of individual factors and their importance in the process of the development of cycling infrastructure, filling the existing research gap. The results of the research can contribute to the discussion on the development of cycle paths by identifying and focusing on key factors. An identical study in another city in Poland could be considered to verify the list of key factors. The authors also consider extending the study to other Central and Eastern European countries, as the list of identified factors is based on the literature and will be applicable to most European countries.

**Author Contributions:** Conceptualisation, A.K.-P., A.T. and G.G.; methodology, G.G.; validation, G.G.; formal analysis, G.G.; investigation, A.K.-P. and A.T.; resources, A.K.-P. and A.T.; data curation, A.K.-P. and A.T.; writing—original draft preparation, A.K.-P., A.T. and G.G.; writing—review and editing, A.K.-P.; visualisation, A.K.-P. and G.G.; project administration, A.K.-P. All authors have read and agreed to the published version of the manuscript.

**Funding:** This research was founded by research subsidies provided by the AGH University of Krakow no. 16.16.150.545 and no. 16.16.200.396.

**Institutional Review Board Statement:** Not applicable.

**Informed Consent Statement:** Not applicable.

**Data Availability Statement:** Data are contained within the article.

**Conflicts of Interest:** The authors declare no conflict of interest.

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
