# Peer review of "Application of the Decision-Making Trial and Evaluation Laboratory Method to Assess Factors Influencing the Development of Cycling Infrastructure in Cities"

_sustainability, doi:10.3390/su152316421_

Round 1
Reviewer 1 Report
Comments and Suggestions for Authors The manuscript focuses on the issue of transportation choices for people to travel, pointing out the important role of bicycle travel as well as the development of bicycle infrastructure in reducing environmental pollution and improving sustainable urban development. And using the DEMATEL methodology to determine the role and prioritization of factors affecting the realization of bicycle infrastructure in Poland as a research object. The results of the study have some practical value for the promotion of green travel modes and the green construction of the city. However, there are still some issues to be explained or revised in the content of the manuscript, for example:
Abstract:
1. There is no need to describe the research process in the abstract, please directly point out the results of the study. Please indicate which factors are key influences on the development of bicycle infrastructure and the causal relationships between the factors.
2. Please conclude with a short statement of the academic value and practical significance of this study.
Introduction:
3. The author only describes the real problems of bicycle infrastructure in Poland, and does not discuss the research related to the construction of bicycle infrastructure, which needs to be deepened in terms of academic innovation.
4. Please ask the scientific questions of this manuscript in direct question form.
Literature review:
5. The logic of the section needs to be improved, as the listing of existing studies is based on a theme or timeline, and the authors have repeated themselves several times in describing the role of bicycle travel in reducing environmental pollution and improving sustainability.
6. When organizing research on the DEMATEL methodology in the transport and infrastructure sector, it is not necessary to refer to other areas, and it is sufficient to stick to the topic of the research.
Materials and Methods:
7. Is a literature search using only the keyword "bicycle lanes" not comprehensive enough?
8. Most of the existing studies combine the DEMATEL method with fuzzy set theory, gray theory, etc. to overcome the shortcomings such as the subjectivity of expert scoring, so why didn't the authors choose a better composite method?
9. In addition, some studies related to the Dematel method were missed.
e.g.,
Govindan, K., Khodaverdi, R., & Vafadarnikjoo, A. (2016). A grey DEMATEL approach to develop third-party logistics provider selection criteria. Industrial Management & Data Systems, 116(4), 690-722.https://doi.org/10.1108/IMDS-05-2015-0180
Gupta, S., Khanna, P., & Soni, U. (2023). Analyzing the interaction of critical success factor for TQM implementation-A grey-DEMATEL approach. Operations Management Research, 1-22.https://doi.org/10.1007/s12063-023-00367-y
Li, X., Li, J., He, J., Huang, Y., Liu, X., Dai, J., & Shen, Q. (2023). What are the key factors of enterprises' greenwashing behaviors under multi-agent interaction? A grey-DEMATEL analysis from Chinese construction materials enterprises. Engineering, Construction and Architectural Management. ahead-of-print. https://doi.org/10.1108/ECAM-01-2023-0027
etc.
Results:
10. It is suggested that the authors rank the influencing factors of A1-A8 according to the calculation results (influence, influenced, cause and center) so that the results can be seen at a glance.
Discussion:
11. The authors have only elaborated the results of the table and the causality diagram without supporting them with existing studies, please add.
In summary, the authors are advised to carefully revise this manuscript in light of the above comments. I sincerely look forward to receiving the revised version.
Author Response
Dear Sir or Madam,
at the beginning we would like to thank You for the review report. Please find below our responses to each of your comments.
Point 1:. Please show the main outcomes of the study in the abstract.
Response 1
The abstract has been corrected.
Point 2: . In the introduction section even literature, you need to add some statements or background related to private vehicle users' motivation, and comparison among other modes. As you know cycling is counted as a private mode. Please see the following resources. Especially in the case of Eastern European Countries.
- a) De Vos, J. Do people travel with their preferred travel mode? Analysing the extent of travel mode dissonance and its effect on travel satisfaction. Transp. Res. Part A Policy Pract. 2018, 117, 261–274.
- b) https://www.mdpi.com/2624-8921/4/2/24 Understanding the Motivation and Satisfaction of Private Vehicle
Users in an Eastern European Country Using Heterogeneity Analysis
- c) Enjoyment of commute: A comparison of different transportation modes Antonio Páez *, Kate Whalen.
Response 2
The adequate information was introduced into the paper.
Point 3: The short literature review presented her proves that DEMATEL comprises a comprehensive and useful tool for dealing with diverse transportation and infrastructural problems. How? please support and justify it strongly, while we have many MCDM techniques. Please see this reference as an example:
https://www.mdpi.com/2076-3417/10/12/4158
Response 3
The literature review was supplemented by necessary explanations and relevant literature.
Point 4: What are the practical implications of this study? not just in Poland but for the world.
Response 4
The adequate information was introduced into the paper.
Point 5: The discussion section is missing in this paper.
Response 5
The discussion has been rewritten.
We hope that the quality of the text has been improved to a satisfactory level.
We look forward to hearing from You.
Yours faithfully,
Authors
Reviewer 2 Report
Comments and Suggestions for Authors
The paper is interesting and well-written. However, the paper needs to be improved considering the following remarks:
1. Please show the main outcomes of the study in the abstract.
2. In the introduction section even literature, you need to add some statements or background related to private vehicle users' motivation, and comparison among other modes. As you know cycling is counted as a private mode. Please see the following resources. Especially in the case of Eastern European Countries.
a)De Vos, J. Do people travel with their preferred travel mode? Analysing the extent of travel mode dissonance and its effect on travel satisfaction. Transp. Res. Part A Policy Pract. 2018, 117, 261–274.
b)https://www.mdpi.com/2624-8921/4/2/24
Understanding the Motivation and Satisfaction of Private Vehicle Users in an Eastern European Country Using Heterogeneity Analysis
c) Enjoyment of commute: A comparison of different transportation modes Antonio Páez *, Kate Whalen.
3. The short literature review presented her proves that DEMATEL comprises a comprehensive and useful tool for dealing with diverse transportation and infrastructural problems. How? please support and justify it strongly, while we have many MCDM techniques. Please see this reference as an example: https://www.mdpi.com/2076-3417/10/12/4158
4. What are the practical implications of this study? not just in Poland but for the world.
5. The discussion section is missing in this paper.
Comments on the Quality of English Language
Minor editing of English language required
Author Response
Dear Sir or Madam,
at the beginning we would like to thank You for the review report. Please find below our responses to each of your comments.
The paper is generally well written and well structured with only minor suggestions for the improvement.
Point 1: Abstract should mention most contributing factors identified in the results.
Response 1
The Introduction section has been corrected and supplemented.
Point 2: In the discussion it should be mentioned whether respondents sample of 18 cyclists and 5 expert is sufficient, and whether larger group of respondents would change the results.
Discussion section is somewhat difficult to follow because it uses factors denotations instead of names and it requires a reader to constantly look in the table. In this regard, I suggest to merge discussion with the conclusions about influencing factors in the conclusion section, and to shorten the conclusion with the brief overview of what was done in the paper.
Response 2
The discussion section was corrected as well as the conclusions were also changed, respectively.
Point 3: A clear distinction of prioritizing factors viewed by bicycle path users and expert should be emphasized and future recommendations provided. Also formatting in the conclusion should be corrected.
Response 3
The adequate information was introduced into the paper.
We hope that the quality of the text has been improved to a satisfactory level.
We look forward to hearing from You.
Yours faithfully,
Authors
Reviewer 3 Report
Comments and Suggestions for Authors
Development of cycling infrastructure has significant impact to sustainable development, environment and human health. However, in many cities where such means of transportation would be beneficial, it is not sufficiently utilized. In this regard, the authors used DEMATEL method to assess the key factors contributing to the development of cycling infrastructure. The paper is generally well written and well structured with only minor suggestions for the improvement.
Abstract should mention most contributing factors identified in the results. In the discussion it should be mentioned whether respondents sample of 18 cyclists and 5 expert is sufficient, and whether larger group of respondents would change the results.
Discussion section is somewhat difficult to follow because it uses factors denotations instead of names and it requires a reader to constantly look in the table. In this regard, I suggest to merge discussion with the conclusions about influencing factors in the conclusion section, and to shorten the conclusion with the brief overview of what was done in the paper. A clear distinction of prioritizing factors viewed by bicycle path users and expert should be emphasized and future recommendations provided. Also formatting in the conclusion should be corrected.
Author Response
Dear Sir or Madam,
at the beginning we would like to thank You for the review report. Please find below our responses to each of your comments.
The manuscript focuses on the issue of transportation choices for people to travel, pointing out the important role of bicycle travel as well as the development of bicycle infrastructure in reducing environmental pollution and improving sustainable urban development. And using the DEMATEL methodology to determine the role and prioritization of factors affecting the realization of bicycle infrastructure in Poland as a research object. The results of the study have some practical value for the promotion of green travel modes and the green construction of the city. However, there are still some issues to be explained or revised in the content of the manuscript, for example:
Point 1: Abstract:
- There is no need to describe the research process in the abstract, please directly point out the results of the study. Please indicate which factors are key influences on the development of bicycle infrastructure and the causal relationships between the factors.
- Please conclude with a short statement of the academic value and practical significance of this study.
Response 1
The abstract has been revised and completed.
Point 2: Introduction:
- The author only describes the real problems of bicycle infrastructure in Poland, and does not discuss the research
related to the construction of bicycle infrastructure, which needs to be deepened in terms of academic innovation.
- Please ask the scientific questions of this manuscript in direct question form.
Response 2
Research related to the construction of bicycle infrastructure in terms of academic innovation has been placed in Chapter 2 Literature Review, (Section 2.1. Issues related to the development of bicycle paths). The introduction was completed by adding academic questions related to the content of the article.
Point 3: Literature review:
- The logic of the section needs to be improved, as the listing of existing studies is based on a theme or timeline, and the authors have repeated themselves several times in describing the role of bicycle travel in reducing environmental pollution and improving sustainability.
- When organizing research on the DEMATEL methodology in the transport and infrastructure sector, it is not necessary to refer to other areas, and it is sufficient to stick to the topic of the research.
Response 3
Literature review has been corrected. Retaining of inclusion of background of the topic of the research was suggested by the other reviewer.
Point 4: Materials and Methods:
- Is a literature search using only the keyword "bicycle lanes" not comprehensive enough?
- Most of the existing studies combine the DEMATEL method with fuzzy set theory, gray theory, etc. to overcome the shortcomings such as the subjectivity of expert scoring, so why didn't the authors choose a better composite method?
- In addition, some studies related to the Dematel method were missed.
- a) Govindan, K., Khodaverdi, R., & Vafadarnikjoo, A. (2016). A grey DEMATEL approach to develop third-party logistics provider selection criteria. Industrial Management & Data Systems, 116(4), 690-722.https://doi.org/10.1108/IMDS-05-2015-0180
- b) Gupta, S., Khanna, P., & Soni, U. (2023). Analyzing the interaction of critical success factor for TQM implementation-A grey-DEMATEL approach. Operations Management Research,
1-22.https://doi.org/10.1007/s12063-023-00367-y - c) Li, X., Li, J., He, J., Huang, Y., Liu, X., Dai, J., & Shen, Q. (2023). What are the key factors of enterprises' greenwashing behaviors under multi-agent interaction? A grey-DEMATEL analysis from Chinese construction materials enterprises. Engineering, Construction and Architectural Management. ahead-of-print.
https://doi.org/10.1108/ECAM-01-2023-0027
Response 4
We strongly agree that there are a lot of literature entries that use non-standard representations of DEMATEL input data. However, we consciously apply crisp DEMATEL technique alternative in the paper. The most important reason for that is that DEMATEL doesn’t need to rely on the application of sophisticated representation of input data, e.g. the fuzzy or grey numbers, to cope with available imperfect (and mostly qualitative) information. That is because DEMATEL is already ready to cover qualitative assessment thanks to the application of a conventional ordinal crisp scale of direct influence. It is also evident from the available literature that the authors of papers which are concerned with the application of non-standard input data representation in DEMATEL aren’t rather concerned about possible negative consequences of using such data e.g. distortion of information caused by the application of arbitrary chosen techniques for additional data processing needed in the case of the application of non-crisp data representation. They don’t even bother with the comparison of final outcomes of the application of their proposals despite the fact that the use of such more ‘advanced’ – data representations – is based on ordinal direct influence scales which may obviously provide no evident change in the actual outcomes, despite more effortful calculations (see: Dytczak and Ginda, 2013) with this regard. Not to mention that the abundance of available imperfect information representations makes the choice of the proper one a cumbersome problem itself. We consciously avoid, therefore, their application in DEMATEL.
We are aware of a huge number of available publications about DEMATEL applications. We nevertheless think that the suggested applications don’t fit well into the scope of our publication and a character of crisp DEMATEL technique applied in the paper.
Point 5: Results:
- It is suggested that the authors rank the influencing factors of A1-A8 according to the calculation results (influence, influenced, cause and center) so that the results can be seen at a glance.
Response 5
We agree with the right suggestion. The respective tables 9 and 13 were introduced to the paper, therefore.
Point 6: Discussion:
- The authors have only elaborated the results of the table and the causality diagram without supporting them with existing studies, please add.
Response 6
We haven’t found similar analyses in the literature. That is why we weren’t able to address anything with this regards.
We revised the manuscript accordingly.
We hope that the quality of the text has been improved to a satisfactory level.
We look forward to hearing from You.
Yours faithfully,
Authors
Round 2
Reviewer 1 Report
Comments and Suggestions for Authors
However, authors didn't revise this manuscript with each review report. I will give you the last chance to revise this manuscript again.
Author Response
Dear Sir or Madam,
at the beginning we would like to thank You for the review report. Please find below our responses to each of your comments.
Please find our response in the attachment.
We hope that the quality of the text has been improved to a satisfactory level.
We look forward to hearing from You.
Yours faithfully,
Authors

Reviewer 2 Report
Comments and Suggestions for Authors
The paper has improved, and thanks for amending the comments.
Still, there are some minor remarks; please revise it again. for example, line 378.
Comments on the Quality of English LanguageIt is fine!
Author Response
Dear Sir or Madam,
Thank you for the kind review and suggestions. We have found them very useful.
In particular, the discussion following line 378 has been expanded.
We hope that the final version of the manuscript is finally acceptable to you.
Yours faithfully,
Authors
Round 3
Reviewer 1 Report
Comments and Suggestions for Authors
Many thanks for your invitation to review this manuscript again. I am pleased that the authors have rewritten this manuscript in good faith and that the current version is acceptable.